# Molecular Characterization of ASFV and Differential Diagnosis of *Erysipelothrix* in ASFV-Infected Pigs in Pig Production Regions in Cameroon

**DOI:** 10.3390/vetsci9080440

**Published:** 2022-08-18

**Authors:** Ebanja Joseph Ebwanga, Stephen Mbigha Ghogomu, Jan Paeshuyse

**Affiliations:** 1Laboratory of Host-Pathogen Interaction in Livestock, Department of Biosystems, Division of Animal and Human Health Engineering, KU Leuven University, 3000 Leuven, Belgium; 2Department of Biochemistry and Molecular Biology, Faculty of Science, University of Buea, Buea P.O. Box 63, Cameroon

**Keywords:** African swine fever, *Erysipelothrix*, genotypes, B646L, E183L, B602L, CP204L, I73R and I329L, genetic and antigenic variability

## Abstract

**Simple Summary:**

African swine fever (ASF, a viral infection) and swine erysipelas (a bacterial infection) are two devastating diseases with similar manifestations causing huge economic losses to the pig industry. Vaccination besides adequate biosecurity is useful in controlling swine erysipelas but not ASF, as there is neither a vaccine nor an antiviral drug against the disease. Our focus was to extensively characterize the ASF virus (ASFV) isolates and to identify the *Erysipelothrix* species circulating in ASFV-infected pigs in Cameroon. Specifically to resolve the intragenomic relatedness (PCR-based genotyping) amongst the ASFV isolates and to differentiate between *E. rhusiopathiae* and *E. tonsillarum* in ASFV-infected pigs. Randomly, we collected 377 samples (blood and tissue) from pig farms and pig slaughter slabs within the major pig production regions. We found that all ASFV isolates belong to Genotype I and contain a single GGAATATATA repeat, that two variants with 19 (ABNAAAACBNABTDBNAFA) and six (ABNAFA) tandem repeat sequences exist, and that 97.30% of isolates belong to serogroup IV. In addition, only *E. tonsillarum* (an avirulent species) and not *E. rhusiopathiae* (virulent species) was detected in all the ASFV-infected samples. Continuous characterisation of the ASFV isolates within Cameroon is necessary for designing effective control measures and future potential ASFV vaccine candidates.

**Abstract:**

African swine fever and swine erysipelas are two devastating diseases with similar manifestations ravaging the domestic pig industry. Only a single phylogenetic study has been carried out in Cameroon, and neither an extensive genotyping aimed at identifying the different serotypes nor has an appropriate differential diagnosis of different species of *Erysipelothrix* has been effected in ASF-infected animals. Of the 377 blood or tissue samples randomly collected from pig farms and slaughter slabs from January to August 2020, 120 were positive for ASFV (by PCR), giving a prevalence of 31.83%. Intragenomic resolution through sequencing divulged the presence of genotypes I, and Ia, two variants with 19 (ABNAAAACBNABTDBNAFA) and six (ABNAFA) tandem repeat sequences (TRS), serotype IV, and a single GGAATATATA repeat. The sole presence of *E. tonsillarum* (avirulent species) and not *E. rhusiopathiae* (virulent species) indicates that the severity observed during the 2020 ASF outbreak in the sampled regions was exclusively due to ASFV genotype I infection. Such characterisations are necessary for designing effective control measures and future potential vaccine candidates.

## 1. Introduction

African swine fever (ASF) is a contagious haemorrhagic swine disease of significant sanitary and socioeconomic importance [1]. The disease, transmitted amongst pigs by the African swine fever virus (ASFV) belongs to the *Asfarviridae* family and affects domestic pigs and wild suids with mortality in domestic and wild boars reaching 100% [2]. First discovered in Kenya in the 1900s, the disease has spread to many other parts of Africa, Europe, and recently China in 2018, leading to a reduction in Chinese pork production by 29.1% [3,4]. Currently, neither vaccines nor antivirals exist, and this has imposed the full implementation of biosecurity measures in many European and South American countries for the eradication of the disease. The viral genome has an isolate-dependent length that ranges from 170–190 kbp. The genome contains 160–175 open reading frames that encode structural proteins besides those necessary for viral replication and survival in the host [2].

Genotyping is usually performed by partial sequence analysis of the C-terminal region of the B646L gene encoding the p72 viral capsid protein [5], and using this method, the ASFV is classified into 24 genotypes. In addition to B646L sequencing, analyses of additional genes have expanded our understanding of the genome diversity and phylogeny of ASFV. As such, intragenomic relatedness amongst isolates that were classified as closely related using partial B646L gene sequences analysis was discovered. This resulted in the identification of genes within the central conserved region of the genome that exhibits tandem repeat sequences (TRS): B602L genes encode a four-amino-acid TRS within the hypervariable central variable region (CVR), the GGATATATA repeats between the intergenic region of I73R and I329L genes, and the hexameric amino acid PPPKPC repeats within the *EP402R* gene encoding the CD2V protein for serotyping, which is hemadsorption inhibition serotype specific and which is vital for vaccine development and E183L (P54 protein) gene for genotyping [6,7,8]. The p54 protein has successfully been used to differentiate isolates from West and East African countries, Europe and America into four distinct genotypes I sub-clusters designated “a”, “b”, “c” and “d” with cluster Ia being the largest and most homogenous group comprising ASFV isolates from Europe and America. Cluster Ib comprises isolates from West African countries, Ic with Lisbon 60a and Mzuki isolates and the East African ASFV isolates of genotypes V, X, and XX into sub-cluster Va, Vb, Xa, Xbm XXa and XXb [6]. Furthermore, the p30 protein (*CP204L* gene) which is the most early expressed and most antigenic ASFV protein located in the cytoplasm of infected cells [9,10,11], has been used to further differentiate among local isolates [7].

The genus *Erysipelothrix* consists of eight species amongst which are the virulent *Erysipelothrix rhusiopathia* and the avirulent *Erysipelothrix tonsillarum* species, which are Gram-positive, non-sporing, rod-shaped, facultative anaerobe bacteria [12]. *Erysipelothrix*
*rhusiopathiae* is of great economic importance, as it causes swine erysipelas in domestic pigs. The disease could either be severe with acute septicaemia resulting in sudden death or less severe to chronic often associated with endocarditis and arthritis. The domestic pig is an important reservoir of *erysipelas*, as 20–40% or in certain cases, 98% of the herd of healthy pigs are carriers of both species in the tonsils [13,14]. To facilitate control and rapid differentiation of erysipelas from ASFV at slaughter slaps, as well as differentiate between the species of *Erysipelothrix,* specific primers for amplification of *Erysipelothrix* DNA have been employed with great success [15,16]. In Cameroon, 73.6% (of the sampled farmers) treat pigs vaccinated against swine erysipelas with antibiotics during ASF outbreaks even though the antibiotics do not affect the virus, and they later sell them to recover some funds [17]. Such bad practices help to convey the antibiotics to humans as the withdrawal periods are not respected. It is, therefore, necessary to investigate the presence *of Erysipelothrix* during ASFV epidemics to (i) ascertain the efficacy of the vaccination against swine erysipelas and (ii) effectively differentiate the species of *Erysipelothrix* when present to establish if severity observed during ASFV epidemics is due to only ASF or both.

Cameroon, with the largest pig population in central Africa (over 3.2 million pigs), employs over 220,000 families [18]. Most of the pig farms are owned by small families with 2 to 10 pigs per farm [17]. Over 70% of the pig farmers are involved in the traditional system of pig production with minimal input (mostly household waste) and biosecurity. Most (80%) of the pigs are hybrids resulting from crosses between exotic breeds or local and exotic breeds [18,19]. The first outbreak of ASF in Cameroon was in 1982, and since then, the country has been endemic with a yearly resurgence of the disease [20,21].

In Cameroon, previously characterized ASFV isolates from 1982 to 2018 belong to Genotype I [21,22] based on B646L gene sequencing while analysis of the E183L gene classified the isolates into subgroups 1a and 1b. Analysis of the central variable region of the *B602L* gene identified three major variants with 19, 20 and 21 tandem repeat sequences (TRS) represented as variants A, B and C, respectively. Variant A, initially present during the first outbreak in 1982, is thought to have mutated to variants B and C by the addition of the “A” representing “CASTˮ within the ABN…….CNB repeat. Variants A and B were distributed throughout the national territory, as opposed to variant C which was found only in Kousseri in the far north region of the country [21]. However, to comprehensively resolve the genetic and serotypes relatedness amongst ASFV isolates in Cameroon, in addition to the *B646L* sequence phylogeny, genetic analysis of the *CP204L*, the intergenic region between *I72L* and *I329L* genes and the *EP402R* genes have been used [6,23,24]. Furthermore, determining the circulating *Erysipelothrix* species which may or may not contribute to the severity of the ASF will help ascertain the efficacy of swine erysipelas vaccination in Cameroon.

In this study, we set out to extensively characterize the ASFV isolates from both local and exotic breeds within four main pig production regions and to particularise the *Erysipelothrix* species (*E. rhusiopathiae* and *E. tonsillarum*) co-occurring with ASFV during the 2020 ASF outbreak in Cameroon. Genotyping of ASFV isolates is of prime importance in epidemiological classification, as it helps in determining specific isolates responsible for outbreaks as well as emerging genotypes due to mutations or introduction from other geographical areas [5]. Due to the endemic nature of the virus in the country, continuous molecular characterisation is vital in monitoring ASFV isolates within the country in order to develop control measures and most probably vaccine strains or subunit vaccines.

## 2. Materials and Methods

### 2.1. Ethical Approval

Ethical approval for this project with code 3E181033 was obtained from the University of Buea Institutional Animal Care and Use Committee (UB-IACUC) on 4 September 2019. Regional authorizations needed for sample collection within specific regions of the country were obtained from the regional delegations of the Northwest, West, Southwest and Littoral regions. Blood was collected only from the pigs of farmers who gave consent for the study. The collection was performed following a standard protocol for jugular vein blood collection in pigs (https://ouv.vt.edu/content/dam/ouv_vt_edu/sops/large-animal/sop-swine-blood-collection.pdf, accessed on 14 November 2020).

### 2.2. Study Area

This study was conducted in the Northwest, Southwest, West and Littoral regions of Cameroon. Representing the main pig production regions in the country, pigs from these regions are either sold within or transported to other regions for a ready market. The ASFV is endemic in these regions with yearly epidemics between April and August.

### 2.3. Study Design

This was a cross-sectional study in which samples (blood and tissue) were collected from pig farms and slaughter slabs. Blood was collected from local, crosses between local and exotics breed, exotic breeds and crosses between exotic breeds from January to March 2020 within pig farms wherein no farmer had declared an ASFV outbreak, whereas sampling at the slaughter slabs was affected during the ASFV outbreak in July 2020.

### 2.4. Sample Collection

Blood (1mL) samples were collected from the jugular veins of apparently healthy or sick pigs in EDTA vacutainer tubes by certified veterinary personnel dressed in disposable biosecurity overalls after informed consent from the pig farmers. All tissue samples were collected into 15 mL falcon tubes containing 1X PBS (Thermo Fisher Scientific, Waltham, MA, USA) supplemented with 1% antibiotic-antimycotic (Thermo Fisher Scientific, Waltham, MA, USA). A total of 377 samples were collected amongst which 277 blood samples were from pig farms in the Northwest (77), West (20), Southwest (143) and Littoral (37) regions [17], while 100 (blood and tissue) were from pig slaughter slabs (46 samples) and the rest (54 samples) were collected in pig farms during the outbreak (Table 1). Blood was collected from pigs aged 2 months and above (weaned), while the tissues (Spleen and lymph nodes) were collected from frozen and newly slaughtered with varied carcass weights ranging from 50 to 200 kg. The mile 16 slab is a private slaughter slab (but open to the public) with 5 to 10 pigs slaughtered per day, while the grand hanger market slaughter slab has about 15 to 20 butchers with 20 to 30 or more pigs slaughtered per day depending on sales (Table 1).

### 2.5. DNA Extraction and ASFV Detection

Genomic DNA was extracted from 300 µL of blood collected in EDTA tubes or 20 mg of infected animal tissue using the Wizard^®^ Genomic DNA Purification Kit (Promega, AJ Leiden, Netherlands) following the manufacturer’s instructions. PCR Amplification of a 257 bp region within the central part of the p72 gene using PPA1 (5’-AGTTATGGGAAACCCGACCC-3’) and PPA2 (5’-CCCTGAATCGGAGCATCCT-3’) primers was performed for ASFV DNA detection for 40 cycles, 94 °C for 2 min, 94 °C for 15 s, 62 °C for 30 s, 72 °C for 30 s and 72 °C for 5 min and a holding step at 4 °C. Amplicons were resolved on a 2% agarose gel and visualised in a Gel Doc imager (Biorad, Hercules, CA, USA).

### 2.6. ASFV Molecular Characterization, Nucleotide Sequencing and Analysis

A series of internationally standardized molecular epidemiology primers were used for genetic and serotype (based on genotyping of the *Ep402R* gene [24]) characterization of the isolates in this study. Primers to amplify the 478bp region of the C-terminal region of the *B646L* gene encoding the p72 protein [5], 675bp fragment of the *E183R* gene encoding the P54 protein [6], a 356bp fragment located between the I73R and I329L region and characterized by GGAATATATA tandem repeat sequence [8,25], a 543bp fragment of the CP204L gene encoding the P30 protein [7,8], a 345bp fragment of the central variable region in the B602L gene [22] and an 816bp fragment of the EP402R gene encoding the cytoplasmic region of the CD2-like protein [24] were used. The PCR conditions were as previously described.

From the PCR product, 5 µL was run on 2% gel to confirm band size and the rest (20 µL) was cleaned using the Wizard^®^ SV Gel and PCR Clean-Up System (Promega, AJ Leiden, Netherlands following the manufacturer’s instruction and sent to Eurofins (Mechelen, Belgium) for Sanger sequencing.

The obtained sequences (with appropriate read lengths) were edited and aligned using CLUSTAL W and Clustal Omega alignment tools within the Bioedit software (https://bioedit.software.informer.com/, accessed on 12 December 2021). MEGA X program (https://www.megasoftware.net/, accessed on 14 December 2021) was used for phylogenetic analysis to construct a neighbour-joining (NJ) tree with 1000 bootstrap replications, and the evolutionary relationship was computed using the P distance method.

### 2.7. DNA Extraction and Erysipelothrix Detection

ASFV-positive samples (120 samples) were subjected to *Erysipelothrix* DNA detection as previously described [13,15]. Primers MO101 (5’-AGATGCCATAGAAACTGGTA-3’) and M0102 (5’-CTGTATCCGCCATAACTA-3’) were used for amplification of a 407 bp fragment of the Erysipelothrix DNA coding sequence of 16S RNA. For differentiation between the *Erysipelothrix* species, primer sets ERY-1F 5’-ATCGATAAAGTGTTATTGGTGG-3’ and ERY-2R 5’-CGAGTGTGAATCCGTCGTCTC-3’ were used to amplify 2210 bp sequences specific for *E. rhusiopathiae* chromosomal DNA, and the primer sets MO101 (5’-AGATGCCCATAGAAACTGGTA-3’) and ERS-1R (5’-CTGATCCGCGATTACTAGCGATTCCG-3’) for amplification of the 719 bp fragment of a highly conserved *Erysipelothrix* DNA that codes for 16S rRNA. Two fragments (2210 and 716 bp) were amplified when *E. rhusiopathiae* alone or both was present, whereas a single fragment (716 bp) was amplified when *E. tonsillarum* was present.

## 3. Results

### 3.1. Diagnosis of ASFV by PCR

A 257 bp fragment of the B646L gene could be amplified by PCR from 120 of the 377 samples, resulting in a prevalence of 31.83%. Of the 277 animals sampled during the pre-outbreak period, 27 were PCR-positive for ASFV while 93 of 100 samples collected during the outbreak sampling were positive. From these 93 ASFV PCR-positive outbreak samples, 50 were from blood collected at farms, and 43 were from tissues/blood collected at slaughter slabs.

### 3.2. B646L (P72) Gene Phylogeny

Characterization of ASFV isolates by comparing the sequences of the C-terminal region of the B646L gene as introduced by Bastos et al., 2006 [5], has been especially useful in epidemiological studies of different ASFV isolates in different countries. Sequences obtained in this study (85 sequences with accession numbers from ON070434–ON070519) were compared with those previously described from Cameroon [5,21], as well as with sequences of other genotypes (genotype I-XXVI) stored in GenBank. The phylogenetic analysis clustered all sequences obtained in this study along with those previously characterized from Cameroon into p72 genotype I. Minor sequence differences were noted with isolates in the pre-outbreak sampling as opposed to the outbreak sampling but were not above the 1% threshold utilized by the MEGA X software to cluster the sequences into different genotypes (Figure 1).

### 3.3. E183L (P54) Gene Phylogeny

The obtained sequences with the expected 658 bp length were aligned with sequences from a study by Wade et al., 2019 (genotypes Ia and Ib) from Cameroon [21] together with sequences from other genotypes. The phylogenetic analysis placed all of the ASFV isolates in this study into genotype Ia (Figure 2).

### 3.4. CP204L (P30) Gene Phylogeny

Amplification of the CP204L gene gave the required 543 bp PCR product. Phylogenetic analysis of the sequence clustered 41.2% of the sequences into genotype I, while 58.8% could not be clustered into any known genotype. Furthermore, 74.1% of the highly polymorphic sequences were from the Misselleleh village in the Southwest region, while the rest (24.9%) were sequences from Douala, Buea, Souza, Limbe and Bafoussam. Pig rearing in this village is largely extensive with local breeds. Although many are crosses between local and exotic breeds, some exotic breeds could be seen in some farms (Figure 3).

### 3.5. Intergenic Region of I73R and I329L

The amplification of the 356 bp fragment between the I17R and I329L genes has been used in the past to differentiate closely related isolates [6]. The major difference is the insertion of the GGAATATATA repeats within these genes. All isolates within this study had a single GGAATATATA irrespective of origin contrary to isolates from Malawi and central Europe with two GGAATATATA repeats and those from Vietnam and Korea with three GGATATATA repeats (Figure 4).

### 3.6. Analysis of the B602L Gene 

Amplification of the hypervariable central variable region (CVR) within the B602L gene was achieved with fragment sizes expected between 300 and 500 bp based on the variant [22]. Following analysis, 21 of the 107 had 19 tandem repeat sequences (TRS) (ABNAAAACBNABTDBNAFA) while 86 of the 107 had six repeats (ABNAFA). All of the 21 samples with 19 repeats were outbreak samples of which seven were collected from Buea, eleven from Douala and two from Misselleleh. The unique pre-outbreak variant with 19 TRS was collected from Misselelleh (Table 2).

Sequence alignment of the variants reveals a NAAAACBNABTDB deletion between the 19 and 6 repeats variants (Figure 5).

### 3.7. Analysis of the EP402R Gene

The 816 bp fragment of the EP402R gene was successfully amplified [24], and the hexameric amino acid repeat PPPKPC was examined. We found a variation in the number of hexameric repeats with 97.30% of the isolates having seven PPPKPC repeats and one each with six and five hexameric PPPKPC amino acid repeats. CMR/Missel_OB33/2020 and CMR/Douala_SL8/2020 had six and five PPPKPC amino acid repeats, respectively (Figure 6).

### 3.8. Differential Diagnosis of Erysipelothrix

For efficient diagnosis and differentiation of the species of *Erysipelothrix *(between *E. rhusiopathiae and E. tonsillarum only)*, 12 of the 120 ASFV positive samples subjected to the PCR-based study were positive for *Erysipelothrix.* All the positive samples were tissue samples comprising four spleen and eight lymph node samples with five collected from the Douala and seven from the Buea slaughter slabs (data not shown). Differential diagnosis revealed the unique presence of *E. tonsillarum* as a unique band corresponding to 719 bp, which was amplified when both primers sets were used or when only MO101 and ERS-1R were used (Figure 7). Except for samples 1 and 6 with either a smear or no clear band, the rest had clear bands corresponding to the 716 bp fragment. 

## 4. Discussion

The persistent epidemic of ASF in Cameroon has led to the scarcity of pigs, thus requiring pig butchers at the Buea and Douala slaughter slabs to travel to other regions within the country to purchase pigs, thereby rendering these slabs representative of pigs from all the 10 regions in Cameroon. Consequently, genotyping of ASFV isolates within these slabs is representative of isolates from the entire country. The results from this study are also representative of the general status of ASFV in Cameroon.

The overall ASF prevalence of 31.83% obtained in this study is lower than the 42.8% prevalence obtained by Wade et al. during an outbreak investigation but is more than the outbreak prevalence of 22.85% obtained by Njayou et al. and 23.85% reported by Victor et al. [21,26,27]. This disparity in prevalence may have been due to differences in sampling period (before or during an outbreak) and the area sampled.

Molecular characterisation of the ASFV isolates by different groups of researchers and at different times placed the isolates from Cameroon into genotype I. The initial study in 1989 by Ekue et al. placed the isolates into group I based on Restriction Fragment Length Polymorphism (RFLP) [28], while genotyping based on the sequencing of the B646L gene placed all isolates from 1989 to 2018 into genotype I [5,21,29]. Genotyping of all isolates within this study placed the isolates into genotype I, which is in line with previous studies [21] (Figure 1).

The local pigs are known to be tolerant to ASF as opposed to the exotic breeds; thus, characterisation of isolates from local and exotic breeds is important in determining genotype diversity. Genotype I is circulating in central Africa with both genotypes I and IX present in the Democratic Republic of Congo. The absence of genotype IX in Cameroon even though there exists cross-boundary trade between the two countries might be a result of efficient border control. These sequences (85) have been deposited in GenBank with accession numbers from ON070434 to ON070519.

Phylogenetic analysis of the E183L gene provides an additional tool to differentiate between closely related ASFV isolates [6,30]. It was successfully used to differentiate isolates from West and East African countries, Europe and America [6]. Phylogenetic analysis clustered all of the isolates from this study into genotype Ia or sub-group Ia when compared with isolates from previous studies in Cameroon and with other isolates with known genotypes (Figure 2).

Our results concur with previous findings by Wade et al., 2018 whose analysis of ASFV isolates from 2011 to 2018 placed all the isolates into genotypes Ia and Ib irrespective of origin [21]. These sequences (86) have been deposited in GenBank with accession numbers from ON191203 to ON191289.

Sequence comparison of CP204L gene fragments of different isolates has successfully been used to differentiate between closely related isolates [8,30]. Analysis of the CP204L gene placed 41.2% (deposited in Genbank with accession numbers ON191340 to ON191380) of the isolates (40) displaying remarkable genetic stability into genotype I, while 58.8% of the isolates displaying remarkable genetic instability did not cluster into any of the other known genotypes (Figure 3).

Interestingly, 74.1% of the sequences displaying remarkable genetic instability were from the Misselleleh village within the Southwest region, while the rest (25.9%) comprised a sequence of isolates from Douala and Souza (Littoral), Limbe and Buea (Southwest) and Bofoussam (West). Furthermore, 95% of the farmers in Misselleleh practise the extensive system of pig production with mostly local breeds and crosses between local and exotic. A greater part (70%) of these pigs are left to fend for themselves with minimal input. These breeds displayed remarkable stamina against the ASFV, as they are asymptomatic and survive during ASFV epidemics in the village as reported by the farmers.

We found a single GGAATATATA repeat upon analysis of the ASFV isolates within this study irrespective of the origin and period of the collection as to pre-outbreak or outbreak. Comparative sequence analysis with isolates with single (Sardinia and Cagliari), double (Malawi and central Europe) and triple (Korea and Vietnam) GGAATATATA repeats revealed a deletion of AA nucleotides at positions 89 and 91 within all isolates with a single GGAATATATA nucleotide repeat but the insertion of T at position 6, TA at position 57 and 58, and GTAGAAAT at position 18 to 24 within isolates with double or triple GGAATATATA nucleotide repeats (Figure 4). These sequences (65) have been deposited in GenBank with accession numbers from ON191381 to ON191446.

The sequencing of the CVR region within this study revealed the presence of two variants with 19 (ABNAAAACBNBTDBNAFA) and six (ABNAFA) TRS with the major difference being deletion of NAAAACBNABTDB in the six TRS as opposed to the 19.

A previous study conducted from 2011 to 2018 found three variants with 19 (ABNAAAACBNBTDBNAFA), 20 (ABNAAAAACBNABTDBNAFA) and 21 (ABNAAAAAACBNABTDBNAFA) TRS repeats named variants A, B and C, respectively. In addition, the initial analysis of the B602L gene of the CAM82 (AAQ08102) isolates by Irusta et al.,1996 revealed the presence of a single variant with 19 (ABNAAAACBNBTDBNAFA) TRS [31]; meanwhile, the analysis of the CAM82 (CAJ90777) by Nix et al., 2006 and Giammarioli et al., 2011 revealed the presence of 23 (ABNAAAACBNABTDBNAAAAANA) TRS [22,32]. Retrieved sequences from Genbank: CAM82 (AAQ08102), CAM82 (CAJ90777), CAM86 (AF513047), CAM89 (AF513045) had 19 (ABNAAAACBNABTDBNAFA), 23 (ABNAAAACBNABTDBNAAAAANA), 20 (ABNAAAAACBNABTDBNAFA) and 20 (ABNAAAACBNAAAACBNAFA) TRS, respectively. As is evident, the 19 and 23 variants were initially present during the 1982 outbreak, but the TRS ABNAAAACBNABTDBNAFA found in 1982, 1996, 2018 and in our study has become the baseline TRS representation with an addition of an “A” representing the “CAST” sequence within the ABNAAAACBN backbone for the 20 and 21 TRS, but the six TRS from our study represent a larger deletion as earlier mention. Evolution of repeated DNA sequence by unequal crossover and slippage synthesis of simple sequence DNA will best explain the appearance of the different variants as well as the large deletion observed in the variant with six TRS. Considering the 19-TRS variant ABNAAAACBNABTDBNAFA, the slippage occurred between the first and last “N” representing the “NVDT” sequence, leading to a deletion of NAAAACBNABTDB in between (Figure 6). The presence of seven variants in Nigeria [29] and two variants in the Democratic Republic of Congo [33] poses a question as to whether border trade between these neighbouring countries has any influence on the variability observed with these variants (Figure 5). These sequences (106) have been deposited in GenBank with accession numbers ON191096 to ON191202.

The EP402R gene or CD2v, similar to Host T cell adhesion molecule CD2, is essential for viral dissemination and is required for hemadsorption of extracellular virion on erythrocyte and red cells around infected macrophages, and its expression in tick cells enhances viral replication [34]. Its proline-rich cytoplasmic domain varies in length between isolates and interacts with the SH3 domain of the SH3P7, an actin-binding adaptor protein involved in vesicle transport, endocytosis and signal transduction within the cell [31,35]. The use of CD2v in serotyping of ASFV isolates was first proposed by Malogolovkin et al., 2004, wherein they showed that the genetic locus encoding the CD2v and C-type lectin proteins of the ASFV mediates HAI serological specificity, rendering it vital in studying ASFV strain diversity in nature [24]. The CD2v and C-type lectin are important but are not sufficient in confirming complete serotype-specific homologous protection and thus the need to discover additional sero-specific antigens [36].To better delineate the isolates within this study, we amplified the EP402R gene and compared the hexameric amino acid PPPKPC repeats within the C terminal region of the gene. Analysis of the repeats revealed the presence of seven PPPKPC amino acid repeats in 97.30% of the samples, while two samples CMR/Missel_OB33/2020 and CMR/Douala_SL8/2020 had six and five repeats, respectively (Figure 6). These sequences (49) have been deposited in GenBank with accession numbers ON191290 to ON191339.

Based on information obtained from the veterinary personnel and the farmers in the sampled regions, the ministry of livestock, fisheries and animal industry (MINEPIA) organizes yearly vaccination against swine erysipelas from January to March, which marks the ASF pre-outbreak period in Cameroon. This is solely the initiative of the different regional delegates’ subsidised by the Cameroon government through MINEPIA [37]. Differential diagnosis of the 12 out of 120 positive samples for *Erysipelothrix* indicates the sole presence of *E. tonsillarum* in our study (Figure 7). The obtained results clearly show that the severity observed during the 2020 outbreak within these regions was solely the effect of the ASFV, which might be indicative of a successful vaccination campaign against swine erysipelas. Further investigations are needed to assess the extent to which the vaccination campaign against swine erysipelas contributes to accomplishing the goal of MINEPIA to eradicate the bacterial infection from Cameroon.

## 5. Conclusions

In-depth genetic and serotype characterisation of the ASFV is important epidemiologically to better apprehend the route of entry and diversity with specific isolates in a country. Here, we concur with previous reports that genotype I remains the sole genotype responsible for the continuous ASFV epidemics in Cameroon. In addition, we found genotype Ia isolate and two variants with 19 and 6 TRS based on analysis of the E183L and B602L genes, respectively. Further intragenomic characterisation from this study revealed the presence of a single GGATATATA repeat, the presence of serotype IV with seven PPPKPC amino acid repeats in 97.30% of the isolates, and the highly polymorphic CP204L gene. The unique presence of *E tonsillarum* within the samples in this study might be indicative of the efficacy of the vaccination against swine erysipelas caused by *E rhusiopathiae*.

Continuous extensive molecular characterisation of ASFV isolates in Cameroon is primordial in designing effective control measures as well as the development of attenuated vaccine candidates necessary for the field situation in Cameroon.

## Figures and Tables

**Figure 1 vetsci-09-00440-f001:**
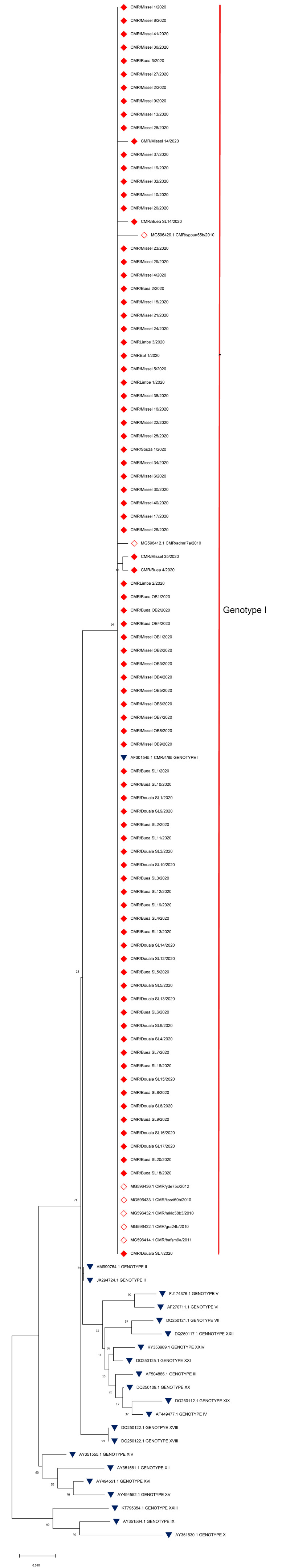
The B646L gene phylogenetic tree constructed using isolates from this study (
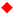
), isolates from previous studies in Cameroon (
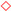
) and other isolates representative of genotypes from GenBank (
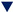
). The evolutionary tree was inferred by the neighbour-joining algorithm method with 1000 bootstraps, evolutionary distance computed using the p-distance method and 85% cut-off.

**Figure 2 vetsci-09-00440-f002:**
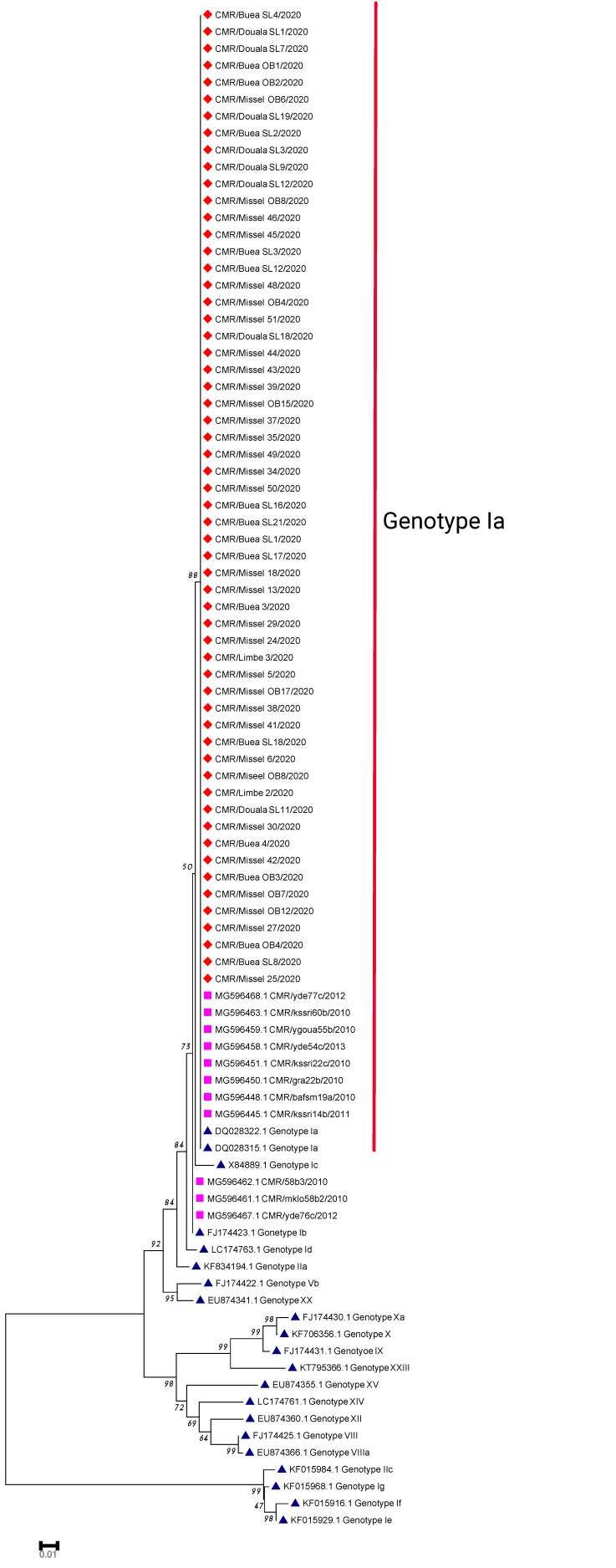
The E183L gene phylogenetic tree constructed using the ASFV isolates from this study, isolates from previous studies in Cameroon and other isolates representative of genotypes from GenBank. The evolutionary tree was inferred by the neighbour-joining algorithm method with 1000 bootstraps, evolutionary distance computed using the p-distance method and 85% cut-off. The samples in this study in sub-group Ia are marked in (
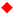
), those from previous studies in Cameroon in (

) and other genotypes from GenBank in (
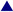
).

**Figure 3 vetsci-09-00440-f003:**
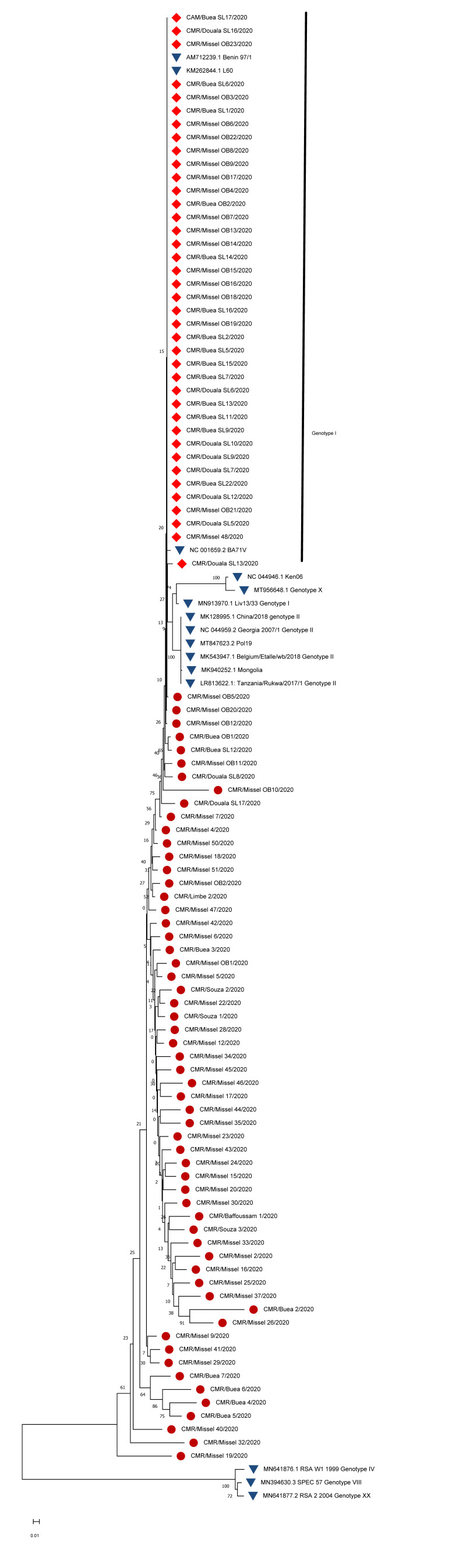
The CP204L gene phylogenetic tree was constructed using the ASFV isolates from this study and other isolates representative of genotypes from GenBank. The evolutionary tree was inferred by the neighbour-joining algorithm method with 1000 bootstraps, evolutionary distance computed using the p-distance method and 85% cut-off. The samples in this study in genotype are marked in (
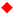
), isolates from this study in no cluster in (
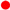
) and other genotypes from GenBank in (
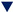
).

**Figure 4 vetsci-09-00440-f004:**
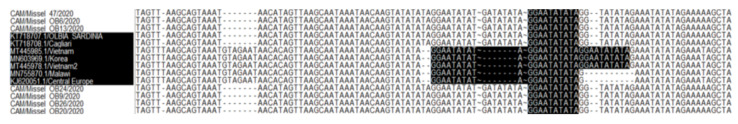
A multiple sequence alignment of the nucleotide sequences of the intergenic region between the I73R and I329L gene from this study with other isolates from Sardinia, Cagliari, Vietnam, Malawi, Korea and central Europe showing the GGAATATATA repeats and other deletions between isolates with single, double and triple GGAATATATA repeats.

**Figure 5 vetsci-09-00440-f005:**
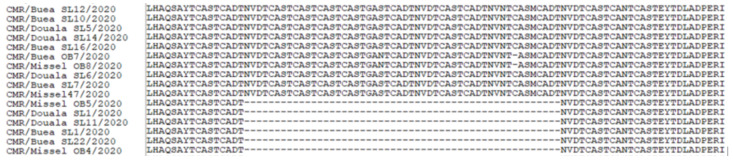
Multiple sequence alignment using Clustal W within the Bio-edit software of the different variants within this study showing the deletion between the 19 repeats variants at the top and the six repeat variants below. The evolution of unequal crossing over and DNA slippage might be useful to uncover the mystery in this variation.

**Figure 6 vetsci-09-00440-f006:**
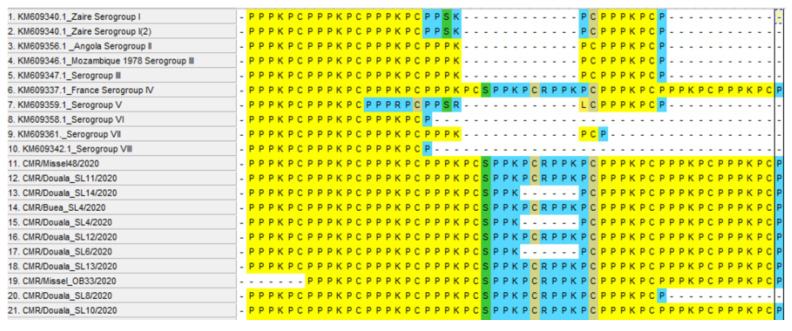
Comparative analysis of the partial EP402R gene of isolates within this study with other isolates with known serotypes. The seven repeats were separated between repeats four and five in 94.60% of the isolates by the SPPKPC and RPPPKC amino acid sequences, while CMR/Douala_SL6/2020, CMR/Douala_SL14/2020 and CMR/Douala_SL4/2020 had the SPPK-------PC thus lacking PCRPPK in between. Furthermore, isolates with the six and five PPPKPC amino acid repeats were both separated between repeats three and three and repeats four and five, respectively, by SPPKPC and RPPPKC amino acid sequence. We compared the isolates with other isolates from the Democratic Republic of Congo, Angola, Mozambique, and France with a determined number of repeats and known serotypes and found the seven PPPKPC amino acid repeats of this study clustered in serotype IV along with the isolate from France, but the five and six repeats did not cluster in any of the known serogroups.

**Figure 7 vetsci-09-00440-f007:**
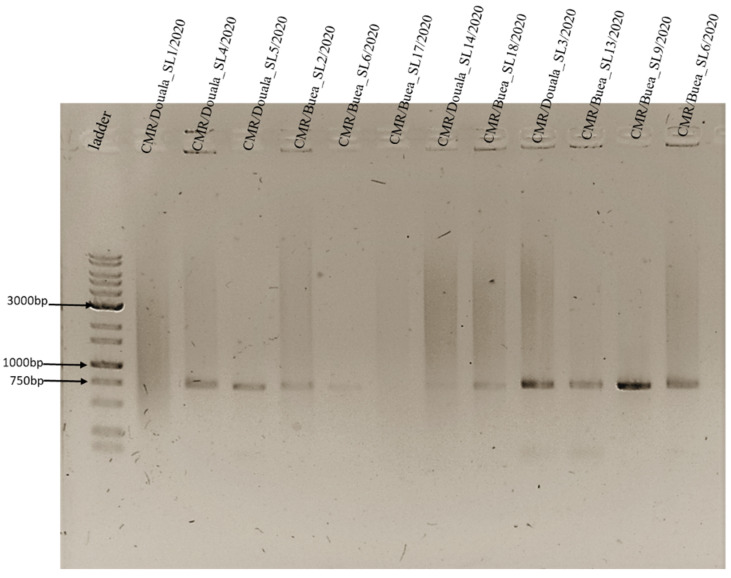
Agarose gel of single 716 bp fragment amplified by PCR for Erysipelothrix differential diagnosis. PCR was performed using primer sets ERY-1F and ERY-2R along with MO101 and ERS-1R in a multiplex reaction. A single 716 bp fragment was amplified for all samples except for CMR/Douala_SL1/2020 and CMR/Buea_SL17/2020 with either a smear or no clear band, respectively.

**Table 1 vetsci-09-00440-t001:** Statistics on regions sampled and the number of samples collected per farm or slab.

Collection Site	Region	Division	Farm Location or Slab	Number of Samples Collected	Total
**Slaughter slabs** **(blood and tissue samples)**	Southwest	Fako	Mile 16 Buea	24	100
	Liottoral	Moungo	Grand Hanger market (Bonaberi)	22	
**Outbreak collection**	Southwest	Fako	`Buea Misselleleh	30 24	
**From pig farms** **(Blood only)**	Littoral	Wouri	Bonaberi	15	37
		Moungo	Souza	22	
	Southwest	Fako	Buea	69	143
			Limbe	26	
			Misselleleh Area	48	
	West	Mifi	Baffousam	8	20
		Koung Khi	Baham	4	
			Bandjoun	8	
	Northwest	Mezam	Bafut	22	77
			Santa	55	

**Table 2 vetsci-09-00440-t002:** Tabular representation of CVR repeats along with accession numbers of sequences involved.

Number of CVR Repeats	CVR Tandem Repeat Sequence (TRS) Signature	Accession Numbers of Sequences Involved
**19**	ABNAAAACBNABTDBNAFA	ON191096, ON191099, ON191100, ON191102, ON191103, ON191104, ON191106, ON191107, ON191110, ON191111, ON191112, ON191113, ON191115, ON191119, ON191121, ON191123, ON191124, ON191127, ON191128, ON191133, ON191134, ON191139, ON191146, ON191155, ON191157, ON191158, ON191160, ON191161,
**6**	AB--------------------NAFA	ON191097, ON191098, ON191101, ON191105, ON191108, ON191109, ON191114, ON191116, ON191117, ON191118, ON191120, ON191122, ON191125, ON191126, ON191129, ON191130, ON191131, ON191132, ON191135, ON191136, ON191137, ON191138, ON191140, ON191141, ON191142, ON191143, ON191144, ON191145, ON191147, ON191148, ON191149, ON191150, ON191151, ON191152, ON191153, ON191154, ON191156, ON191159, ON191162, ON191163, ON191164, ON191165, ON191166, ON191167, ON191168, ON191169, ON191170, ON191171, ON191172, ON191173, ON191174, ON191175, ON191176, ON191177, ON191178, ON191179, ON191180, ON191181, ON191182, ON191183, ON191184, ON191185, ON191186, ON191187, ON191188, ON191189, ON191190, ON191191, ON191192, ON191193, ON191194, ON191195, ON191196, ON191197, ON191198, ON191199, ON191200, ON191201, ON191202.

## Data Availability

Sequences used in this study have been deposited in GenBank, and any other data will be provided upon request.

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
