# Peer review of "Molecular Characterization of ASFV and Differential Diagnosis of Erysipelothrix in ASFV-Infected Pigs in Pig Production Regions in Cameroon"

_vetsci, 2022, doi:10.3390/vetsci9080440_

Round 1
Reviewer 1 Report
The paper “Molecular characterization of ASFV and differential diagnosis of Erysipelothrix in ASFV infected pigs in pig production regions in Cameroon” provides some results on the occurrence of ASFV and swine erysipelas in Cameroon.
However, the Authors do not provide some important information for the interpretation of the results. Furthermore, some conclusions do not have, in the opinion of the Reviewer, a sufficient scientific basis.
Considering the current epidemiological situation of ASF, any information is valuable. However, it is my opinion that the manuscript needs significant modifications to be published.
As far as the content is concerned, I suggest the authors to take into account the following comments:
- Please carefully read the instructions for authors and use the template correctly
- In several parts of the text: “Erysipelothrix rhusiopathiae” not “Erysipelothrix rhupsiopathiae”
- make erysipelas (italics) uniform in erysipelas (not italics) throughout the text
L. 1. Indicate the type of paper (Article)
L. 4. Delete “.” after “Cameroon”
L. 17. Add “.” after “industry”
L. 21. “till” or “still”?
L.15-29. The Simple Summary must be rewritten, indicating more clearly: the aims and objectives of the work, the type and number of samples examined, the results obtained. Furthermore, the authors indicate (here and in other parts of the text) E. tonsillarum as an avirulent strain (it would be better to indicate E. tonsillarum as a species). However, strains of E. tonsillarum have been isolated from cases of chronic arthritis and vegetative valvular endocarditis (Bender et al. 2011; Takahashi et al. 1996).
L. 36. “33.05%". See my comments at L. 167-168.
Introduction. To allow the reader to better interpret the results, the Authors should provide some information on the characteristics of Cameroon pig farms, quantity of pigs present, number of farms, etc. See also my comments at L. 124
L. 50-52. The sentence must be rewritten specifying that the natural hosts of ASFV are wild and domestic suids. European wild boars are susceptible to ASFV infection and present clinical signs and mortality rates similar to those observed in domestic pigs and that ASFV usually produces subclinical infection in the African wild suid species. “Asfarviridae” not “asfarviridae”.
L. 53. china/China
L. 58 and in other parts of the text. Add a space after the number. e.g. “170 – 190 Kbp” NOT “170 – 190Kbp”
L. 82-84. The genus Erysipelothrix comprises seven species to date
L.90 and 93. change ASFV to ASF
L. 115-123. This paragraph must be rewritten clearly indicating the aims of the work
L. 124. A subparagraph must be added (eg Samples or Animal or Study area). I recommend to the authors to use the one written in their previous work (https://doi.org/10.3390/agriculture12010044) as a scheme.
It is my opinion that the reader should find in this paragraph all the information on the samples examined without being forced to read another paper. This paragraph must provide information on: type of farm, number of animals raised, their weight/age at slaughter, etc. Information on the age of the sampled animals, number of samples taken, methods of collection, conditions of transport of the samples, etc. must also be provided. Information about the slaughterhouses (pig slaughter only?, number of pigs slaughtered / day, etc.) should also be provided.
L. 125. The first sentence is incomplete.
L. 167-168. If 120 out of the 377 samples tested positive, the prevalence is 31.83% (120/377) not 33.05%.
L. 178-179. Indicate the number of sequences and their selection criteria
L. 209. ALL/All
L. 210-211. I guess Buea, Douala, Misselleleh are three towns / villages. See my comment to L. 124 on the description of the study area.
L. 213. Tables should be inserted into the main text close to their first citation
L. 220. Delete “.” after Erysipelothrix
L. 221. Specify that the differentiation was made only between two species of Erysipelothrix
L.223. “nodes”. I guess the authors mean lymph nodes. Which?
L. 231. “Figure 1” is repeated twice
L. 235. General comment. In the Discussion section, many results are reported that should instead be included in the Results section
L. 239-241. The Authors say the study results are representative of the ASVF status in Cameroon. The Reviewer believes that in the absence of valid sampling criteria (e.g. proportion of animals tested stratified by their geographic origin) this statement has no valid scientific basis.
L. 242. See my comment to L. 167-168.
L. 252. Add references
L. 255. “Figure 2” is repeated twice. Figures 2, 3 and 4 have a low resolution.
L. 289. “Figure 4” is repeated twice.
L. 385. Figure 7: Figure 8:
L. 401-402. I suggest more caution. It is my opinion that the number of samples examined does not allow to consider the swine erysipelas eradicated at national level.
L. 413-414. See previous comment
L. 427-429. Indicate what is required or delete the paragraph
Author Response
Responds to the comment attached

Reviewer 2 Report
al.
This study provides new evidence for molecular characterization of ASFV and differential diagnosis of Erysipelothrix in ASFV infected pigs. Generally, the current article has a serious lack of evidence study design, such as about sampling from pigs.
Comments to the authors
§ L115-117: please give more details in materials and methods
§ L124-132: 2.1 to 2.5 sections are missing
§ L124-125: give details about sampling from both local and exotic breeds within four main pig production Regions in Cameroon, as you referred in L115-117
§ L206-213; add the appropriate references
§ L215-219: add the appropriate references
§ L251-253, L270-271, L286-287, L319, L361-364, L384-392: remove the figures and tables in part of ‘’results’’
Author Response
Responds to the comments attached

Round 2
Reviewer 1 Report
Dear Authors, it is my opinion that you have answered my questions.
The manuscript has significantly improved. However, I still have some minor concerns:
- L. 26: E. rhusiopathiae is virulent
- L. 81: “Erysipelothrix rhusiopathiae” not “Erysipelothrix rhusiopathia”
- L. 122: “E. tonsillarum” not “E tosillarum”
Author Response
All adjustments have been made (attached).

Reviewer 2 Report
Dear editor
I accept the corrections in the revised manuscript.
I propose to accept this article for publication.
Author Response
No comments from reviewer two
